# OpenReview forum: "Multilingual Unlearning in LLMs: Transfer, Dynamics, and Reversibility"
_ICML.cc/2026/Conference — ICML 2026 regular_

### Official Review · Reviewer_bDeP · 2026-03-11

**Soundness:** 4
**Presentation:** 3
**Significance:** 4
**Originality:** 3
**Overall Recommendation:** 5
**Confidence:** 4

**Summary:**

Unlearning removes targeted knowledge in Large Language Models (LLMs), but previous work mainly focuses on monolingual dynamics. The paper performs a systemic study of the dynamics of unlearning based on parameter-update based methods across many languages, models, and settings uncovering noteworthy patterns regarding the importance of sharing scripts and language families for unlearning transfer between languages and the layer-wise dynamics of multilingual unlearning. The paper then further investigates these patterns via mechanistic interventions with steering vectors to support the hypothesis that parameter-based unlearning tends to act as suppression rather than true knowledge erasure with implications for future unlearning method evaluation and vulnerability considerations.

**Compliance With Llm Reviewing Policy:**

Affirmed.

**Key Questions For Authors:**

1. Have you tried any direct analysis of the steering vectors to see if they encode any patterns or trends across layers and languages?
2. Why was English chosen as the language to compute steering vectors in, why not other languages?
3. Have you tried analyzing non-synthetic settings where you try analyzing real-world data?

**Limitations:**

The paper overall discusses the limitations of the work well, except for the fact that the discussion should include more discussion about how the scope of this work is limited to knowledge gained during SFT only and not during pretraining.

**Strengths And Weaknesses:**

**Strengths:**
- The paper is well motivated, explores a timely issue, and has potential real-world implications on LLM safety.
- The paper is extremely well-written and presents a convincing narrative.
- The methodology is sound and very thorough and the paper supports most of its major claims (while avoiding over-claiming for a majority of the discussion) with both observational and experimental evidence.
- The layer-wise findings and use of mechanistic interpretability to diagnose and (highlight the deficiencies of) the dynamics of multilingual unlearning is novel as far as I know.

**Weaknesses:**
- A more in-depth analysis of the actual concept directions of the steering vectors across languages is missing, which keeps the claims of the paper that “the [steering] vectors capture a general unlearning direction” (section 4.4 L409-412) with relatively less concrete evidence given that the interventions where only sourced from English. Although the English vectors did indeed have ripple effects on the other languages and this is compelling evidence, it is not enough to suggest the existence of a language-agnostic direction/subspace of unlearning. This can be verified by specifically analyzing the steering vectors constructed from multiple languages and seeing if they share some direction/subspace.
- The work only explores unlearning dynamics of knowledge gained during SFT only. Existing work [1] shows that knowledge editing of information acquired during pretraining and after SFT exhibits differing behavior, and hence it puts doubt on whether the same findings hold on unlearning of pretrained information and limits the scope of the findings to the SFT case. An exploration on custom models or open pretreaining data models to explore the exact differences between multilingual unlearning at the pretraining and at the SFT stage would allow for deeper insights. More discussion on the limitations of the scope of the findings of the paper would help elucidate the positioning of the paper’s contribution.

*Minor comment on presentation:*
- The paper’s organization of the figures can be improved so that they are seen closer to where they are referenced in the text (i.e. Table 1, Table 2, Figure 1)

[1] Anna, B., Savchenko, A., Panchecko, A., & Tutubalina, E. (2026). Anatomy of Unlearning: The Dual Impact of Fact Salience and Model Fine-Tuning. arXiv preprint arXiv:2602.19612.

---

> ### Author Rebuttal · Authors · 2026-03-31
>
> Thank you for the positive assessment of our work and the useful feedback, which we address below. We will be very happy to engage with follow-up questions in the discussion phase.
>
> **Weakness 1 and Question2**: Ripple effects of English-based steering vectors and “language-agnostic direction”.
>
> **Response**: When constructing the steering vector, we use identical inputs for the auxiliary unlearned model and the fine-tuned model, so the resulting vector is not attributable to differences in semantic input (Lines 372–377). As such, the observed effect cannot be driven by the semantics of a particular source language. In this sense, we used “language-agnostic direction” to describe the observed cross-lingual generalization of the intervention. We will revise the wording in the paper to make this claim more precise.
>
> To further demonstrate that the steering effect is largely independent of the source-language choice, we repeated the steering vector experiments from the paper using Chinese as the source language. We obtain an average recovery of 51% forget knowledge, with results ranging from 40% for Russian to 68% for English, with the strongest effect around layer 23. These results are qualitatively consistent with the findings reported in Figures 2 of the paper. Due to time constraints, we could not finish all four languages by the response deadline but can add them in the discussion period if needed.
>
> **Weakness 2**: The work only explores unlearning dynamics of knowledge gained during SFT.
>
> **Response**: This is correct; however, the SFT setting is especially relevant in practice, because developers often adapt base models to downstream applications by fine-tuning them on private or proprietary data. As a result, fact-level unlearning after SFT is a common and practically important scenario. We will make our contribution more explicit in the paper and incorporate the reference you provided to further support this discussion.
>
>
> **Other questions**: Steering vector analysis across layers and languages and real-world data
>
> **Response**:  For part 1, we have not conducted a direct analysis of the steering vectors themselves, because the focus of this paper is multilingual unlearning, and our primary goal is to test whether the steering intervention induces consistent cross-lingual behavioral effects. A vector analysis would certainly be valuable, but it addresses a related yet distinct question about the mechanism of the intervention rather than its role in multilingual unlearning. We therefore view this as beyond the scope of the current paper.
>
> Regarding part 2, we have not evaluated the method in non-synthetic settings in the current paper. A central challenge is identifying target data that can be confidently treated as unseen by the model. Even benchmarks released after the model may still partially overlap with pretraining data, making it difficult to establish a clean evaluation setting with confidence [1,2,3]. That said, our current setting is still of high practical relevance: LLMs may inadvertently memorize factual information, such as personally identification numbers or phone numbers, that may later need to be unlearned. Our data and experimental setup simulates such a scenario.
>
> [1] https://arxiv.org/pdf/2406.07933
>
> [2] https://arxiv.org/pdf/2310.16789
>
> [3] https://arxiv.org/pdf/2310.18018

---

> > ### Author Rebuttal · Reviewer_bDeP · 2026-04-02
> >
> > I maintain my positive rating.

---

### Official Review · Reviewer_x46o · 2026-03-12

**Soundness:** 2
**Presentation:** 3
**Significance:** 3
**Originality:** 2
**Overall Recommendation:** 4
**Confidence:** 3

**Summary:**

This paper presents the first comprehensive study of machine unlearning in multilingual large language models (LLMs). The authors extend the TOFU benchmark to four additional languages (Chinese, German, Russian, Turkish), creating a controlled setting to evaluate cross-lingual transfer of unlearning. Through systematic experiments varying fine-tuning, unlearning, and query languages, they demonstrate that unlearning transfer is highly variable: it is strongest between typologically similar languages sharing scripts and families, and exhibits asymmetry favoring high-resource languages as unlearning sources. Mechanistically, the paper reveals that unlearning primarily operates on late decoding layers while leaving early and middle cross-lingual representation spaces largely intact, suggesting that knowledge is suppressed rather than erased. Leveraging this insight, the authors extract "unlearning directions" (steering vectors) from activation differences and show that a single inference-time intervention can recover 50-90% of supposedly unlearned knowledge across languages without additional training or access to the original forget data.

**Compliance With Llm Reviewing Policy:**

Affirmed.

**Final Justification:**

Please see the rebuttal acknowledgement.

**Key Questions For Authors:**

NA

**Strengths And Weaknesses:**

Strengths：
1.This paper is technically rigorous and well-executed.
2.This work advances understanding by connecting behavioral cross-lingual transfer to internal representational geometry, offering a mechanistic framework for evaluating unlearning robustness. The multilingual TOFU extension provides a valuable resource for future research.
3.The work offers novel insights by systematically characterizing multilingual unlearning transfer for the first time, revealing script and family dependencies that were previously anecdotal.

Weaknesses：
1.While the evidence for suppression is compelling, the claim that knowledge is "not truly erased" relies on the interpretability of steering vectors as reversing suppression rather than adding back specific information—a subtle distinction that could be discussed more carefully.
2.While the specific application is novel, the steering vector methodology itself builds heavily on existing activation steering literature. The paper could more clearly articulate how the extraction of "unlearning directions" differs from standard contrastive activation addition.
3.The observation that high-resource languages dominate cross-lingual transfer aligns with established findings in multilingual learning (Lin et al., 2019), though the application to unlearning is new.

---

> ### Author Rebuttal · Authors · 2026-03-31
>
> Thank you for the positive assessment, and thorough feedback, which we address below. We will be very happy to engage with follow-up questions in the discussion phase.
>
> **Weakness 1**: Steering vector suppression vs. “true erasure” of knowledge.
>
> **Response**:  We would like to clarify a potential misunderstanding of our steering setup. The steering vector is computed from the hidden-state difference between the fine-tuned model and an auxiliary unlearned model under identical inputs, using an auxiliary dataset that does not overlap with the forget examples used for evaluation. Therefore, the steering-based analysis does not leak information from the forget set. For this reason, if applying the vector enables the model to recover the forgotten answer, we interpret this as more consistent with suppression, i.e., knowledge remaining in the model.
>
> More broadly, our conclusion that knowledge is merely suppressed does not rest on steering alone. The cosine-similarity analysis shows that the unlearned model remains close to the fine-tuned model through the early and middle layers, with divergence emerging mainly in the later decoding layers (Figure 1). In addition, the PCA and centroid-distance analysis shows that the unlearned representations remain closer to the fine-tuned model than to the base model (Figure 5). Taken together, these findings are more consistent with suppression or reorganization than with complete erasure. We will clarify our reasoning in the paper. Please let us know whether  this clarifies your concern or if we can provide any further explanations.
>
> **Weakness 2**: While the specific application is novel, the steering vector methodology itself builds heavily on existing activation steering literature. The paper could more clearly articulate how the extraction of "unlearning directions" differs from standard contrastive activation addition.
>
> **Response**: We use steering as an analysis tool to probe the internal effect of unlearning. This differs from standard contrastive activation addition, which usually derives directions from semantic contrasts between different prompts or attributes in order to induce a target behavior. In contrast, our use of steering is intended to probe the representational shift caused by unlearning. Using this setup, we find additional evidence that the unlearning effect is better characterized as suppression than complete erasure, which in turn has potential privacy implications. We also find that this suppression effect is language-agnostic,i.e., knowledge in all languages can be restored from a single EN-based steering vector. Beyond this analysis, we believe our proposed steering-based probing method will provide a useful robustness test for future unlearning methods. We will revise the paper to make this distinction from prior steering work more explicit.
>
> **Weakness 3**: The observation that high-resource languages dominate cross-lingual transfer aligns with established findings in multilingual learning (Lin et al., 2019), though the application to unlearning is new.
>
> **Response**:  Our contribution is not the asymmetry itself, which indeed has been shown before. Instead, to the best of our knowledge, we are the first to identify and empirically demonstrate its very practical real-world impacts through privacy implications in multilingual LLMs. Specifically, because unlearning transfers less effectively to low-resource languages, privacy protection is disparate, leaving those languages more vulnerable to cross-lingual attacks. Beyond this observation, our paper also contributes a novel analysis of the internal dynamics of multilingual unlearning and shows that steering-based intervention is a powerful tool for probing and testing unlearning behavior.

---

> > ### Author Rebuttal · Reviewer_x46o · 2026-04-05
> >
> > NA

---

### Official Review · Reviewer_gfMr · 2026-03-12

**Soundness:** 2
**Presentation:** 3
**Significance:** 3
**Originality:** 2
**Overall Recommendation:** 2
**Confidence:** 3

**Summary:**

This paper points out an important and critical issue: how does the current unlearning paradigm perform in multilingual scenarios? The authors mainly focus on three aspects: 1. whether the suppressed content is recoverable in multilingual models; 2. whether the effect is language-specific or language-agnostic; 3. where in the model’s representation space it is localized. The authors develop the benchmark to five different languages and conduct comprehensive experiments across different settings and models. The results show that cross-lingual unlearning effects are highly correlated with the language family. In particular, the authors propose that unlearning does not actually eliminate knowledge, but rather causes superficial suppression by acting primarily on the later decoding layers of the model.

**Compliance With Llm Reviewing Policy:**

Affirmed.

**Final Justification:**

The author's response addresses my concerns, but considering that getting to the final version may require major revisions, I decide to maintain my score.

**Key Questions For Authors:**

Please refer to the above weaknesses.

**Limitations:**

Please refer to the above weaknesses.

**Strengths And Weaknesses:**

### **Strengths**
- This paper focuses on a very important and meaningful issue.
- The authors not only provide valuable empirical results, but also conduct in-depth analyses based on language families and the model's latent representations.
### **Weaknesses**
- Only DPO-style unlearning method used in this paper. In order to verify the generalizability of the conclusions in this paper, experiments on other unlearning methods are also very important. In general, there are many differences between the characteristics of different unlearning methods.
- The multilingual extension of the original English dataset is obtained based on Gemini-flash2.5 in this paper. However, some of the "cross-lingual shared space" observed in the paper may be partly the result of the high degree of normalization of the parallel translation data, and not entirely the result of the model's natural multilingual organization. The paper does not give a sufficiently adequate analysis of translation itself.
- A single distribution of experimental data limits conclusions to specific conditions.
- The authors omit the results on the retain set in the main text and mention that there are small changes on the retain set. However, the core of the unlearning method still lies primarily in the trade-off between forget quality and model utility. Without demonstrating the model utility, it will be difficult for readers to determine whether the method locally destroys the model's capabilities when performing forgetting.

---

> ### Author Rebuttal · Authors · 2026-03-31
>
> **Weakness 1**: Generalization beyond DPO-style unlearning.
>
> **Response**:To test the generalizability of our conclusions, we additionally evaluate two widely used unlearning methods, Negative Preference Optimization (NPO) and Gradient Ascent (GA), using Gemma 2 under the same experimental setup as in the paper. Both methods show patterns consistent with those observed under DPO unlearning, providing evidence that our conclusions hold across diverse and widely-used unlearning methods. Due to the rebuttal limit, we report partial results for cross-lingual transfer (Table 1) and steering vectors (Figure 2) below, and would be happy to provide the full results during the discussion stage if helpful. This table is interpreted in the same way as Table 1 in the paper.
>
> Cross-lingual Transfer pattern (NPO):
> |Family|Unlearn|Query-EN|Query-CH|Query-DE|Query-RU|Query-TU|
> |---|---|---|---|---|---|---|
> |FT-En|Base|97|12|86|47|31|
> ||EN|-85|-6|-73|-34|-19|
> ||CH|-28|-4|-30|-22|-10|
> ||DE|-35|18|-50|-17|0|
> ||RU|-35|10|-36|-28|3|
> ||TU|-46|11|-38|-21|-21|
> |FT-Tu|Base|19|6|18|15|91|
> ||EN|-14|1|-5|-5|-60|
> ||CH|10|0|28|2|-20|
> ||DE|-6|15|-13|15|-26|
> ||RU|1|21|24|-10|-12|
> ||TU|-5|14|7|10|-77|
>
> GA:
> |Family|Unlearn|Query-EN|Query-CH|Query-DE|Query-RU|Query-TU|
> |---|---|---|---|---|---|---|
> |FT-En|Base|97|12|86|47|31|
> ||EN|-92|-6|-68|-32|-22|
> ||CH|-35|-2|-40|-18|-8|
> ||DE|-61|14|-48|-10|0|
> ||RU|-68|15|-55|-22|-13|
> ||TU|-46|6|-46|-15|-15|
> |FT-Tu|Base|19|6|18|15|91|
> ||EN|-3|8|-9|-1|-79|
> ||CH|25|0|16|0|-42|
> ||DE|0|12|-7|3|-74|
> ||RU|8|13|8|-2|-64|
> ||TU|-3|24|8|29|-69|
>
> **Steering Vectors**:
> For NPO, the steering vector recovers 61% of the forgotten knowledge on average (Min=51% Turkish; max=69% English), and is most effective around layer 26 across languages. For GA, average recovery is 59% (min=42% German; max= 81% Chinese), with the strongest effect around layer 20. These results are consistent with the DPO findings reported in Figures 2 and 8 of the paper. Thank you for this valuable suggestion. Evaluating additional unlearning methods further strengthens the credibility of our findings, and we will include complete results in the revision.
>
> **Weakness 2**:Gemini translations might over-estimate the cross-lingual shared space / add translation analysis.
>
> **Response**:We would like to point out that the high degree of normalization across languages in our dataset was an intentional design choice: we aimed to keep semantic content as closely aligned as possible across languages so that differences in unlearning behavior could be compared under controlled conditions, rather than being confounded by differences in wording or linguistic complexity. Without this control, it would be difficult to determine whether variation in unlearning transfer arises from language-specific representations or merely from translation-induced input differences. Our experimental setting is also practically relevant: it emulates a situation where the exact same piece of sensitive information appears in multiple languages in highly parallel form, such as phone numbers or personal ID numbers occurring in multilingual documents. We will clarify this in the paper.
>
> **Weakness 3**:A single distribution of experimental data limits conclusions to specific conditions.
>
> **Response**: Our goal is to study a controlled setting of small-scale forgetting of specific factual knowledge. We focus on TOFU because, in unlearning, it is difficult to identify target data that can be confidently treated as unseen by the model. Even datasets released after the model may still overlap with its pretraining data, making the effects of fine-tuning and unlearning difficult to interpret [1,2,3]. If the model has already encountered the target facts during pretraining, the resulting behavior may reflect prior memorization rather than the intended unlearning process. We will clarify this in our paper.
>
> [1] https://arxiv.org/pdf/2406.07933 [2] https://arxiv.org/pdf/2310.16789 [3] https://arxiv.org/pdf/2310.18018
>
> **Weakness 4**:Add retain set performance
>
> **Response**: Thank you for this point, while we promised to release these results in a repository (paper footnote 5), we agree that they should be in the paper. We will add these results to the Appendix and provide results for Qwen below (truncated due to character limitations). As in Table 1, each cell reports the absolute change in retain-set NLI score after unlearning in the corresponding language, relative to the base model. We omit CIs, which are very low and comparable to Table 1 in the paper. We will be happy to provide additional results during the discussion stage if helpful.
>
> |Family|Unlearn|Query-EN|Query-CH|Query-DE|Query-RU|Query-TU|
> |---|---|---|---|---|---|---|
> |FT-EN|Base|92|9|17|12|8|
> ||EN|-7|0|-1|-1|-1|
> ||CH|-1|-1|0|0|0|
> ||DE|-1|0|2|0|0|
> ||RU|-1|-1|-1|-2|0|
> ||TU|-1|0|0|0|-2|
> |FT-TU|Base|11|9|10|10|93|
> ||EN|-7|-1|-1|-2|-2|
> ||CH|-2|-3|0|-2|-2|
> ||DE|-1|0|-2|-2|-2|
> ||RU|-2|-1|-1|-6|-2|
> ||TU|-1|0|0|-1|-11|

---

> > ### Author Rebuttal · Reviewer_gfMr · 2026-04-03
> >
> > Thank you for your reply. The author's response addresses my concerns, but considering that getting to the final version may require major revisions, I decide to maintain my score.

---

> > > ### Author Response · Authors · 2026-04-04
> > >
> > > Weakness 1 was addressed through additional experiments, which are fully complete, including all results. These results are consistent with, and fit well within, the paper's existing discussion and conclusions regarding the generalization of unlearning algorithms (Lines 421–426). Incorporating them into the paper would therefore constitute a minor revision and is entirely feasible within the extra page.
> > >
> > > Weaknesses 2 and 3 do not require revisions beyond the short explanations already provided in our response.
> > >
> > > The results for Weakness 4 already existed and were promised for release in the paper (Footnote 5). They will be pasted into the appendix, with a prominent reference from the main paper with minimal effort.
> > >
> > > We hope this clarifies the extent of the revisions, and addresses your concerns. Thank you for your engagement with our paper.

---

### Official Review · Reviewer_mtH3 · 2026-03-15

**Soundness:** 2
**Presentation:** 3
**Significance:** 2
**Originality:** 2
**Overall Recommendation:** 3
**Confidence:** 3

**Summary:**

This paper investigates the multilingual configurations of LLM unlearning across five languages: English, Chinese, German, Russian, and Turkish. In detail, only one dataset (TOFU) and one algorithm (DPO), applied to two LLMs (Gemma and Qwen), are considered for evaluation.

The experimental results show that the prevailing problems of unlearning algorithms are also pervasive in multilingual settings, including the fact that the algorithm tends to suppress knowledge rather than truly remove it. Furthermore, similar to typical cross-lingual transfer scenarios, the effectiveness of unlearning against multilingual attacks depends on the languages used, particularly on how similar the languages are.

Additional analysis of internal representations shows that performance gaps across languages arise primarily from decoding bottlenecks.
The authors also demonstrate that, by introducing vector steering, unlearned knowledge can be easily recovered in multilingual settings.

**Compliance With Llm Reviewing Policy:**

Affirmed.

**Key Questions For Authors:**

- Recent work in machine learning (LLMs) has reported that different unlearning approaches exhibit different characteristics, strengths, and weaknesses. Considering this, can the findings of this paper be generalized to other methods? If so, why? If not, what is a potential remedy for this work?
- Are the models used in the experiments (Qwen2.5-7B and Gemma2-9B) truly multilingual? Although most current LLMs are typically considered multilingual, their performance on multilingual tasks can vary significantly depending on model design and pre-training strategies. Verifying the multilingual competence of these models, or introducing other models that are purposely designed for multilingual tasks, could strengthen the claims of the paper.

**Limitations:**

- (Minor) Please consider incorporating some visualizations to better convey the core concepts of this work.

**Strengths And Weaknesses:**

Strengths
- This paper focuses on multilingual settings for unlearning, which have received relatively little attention in the literature.
- Experimental results confirm that problems observed in monolingual settings also prevail in multilingual settings, potentially being even more severe due to the more complex nature of multilingual tasks.

Weaknesses
- As already stated in the paper, caution is warranted when attempting to generalize the findings of this work because it only tests one algorithm on one dataset (TOFU). These limited experimental configurations severely restrict the generalization of the reported results.
- There are no details on the quality validation of the translation (or data generation) process conducted in the paper. Moreover, the evaluation utilizes an NLI-based semantic score, which differs from typical evaluation choices, and its effectiveness needs to be verified. I personally do not believe that the performance of NLI models based on XLM-RoBERTa-large is reliable enough to be used as a sole metric. Introducing auxiliary metrics or additional evaluation strategies would be appreciated. The human validation of 50 TOFU samples deserves credit; however, this approach may not be sufficiently generalizable or scalable.
- The finding that language similarity affects cross-lingual transfer is not particularly new in the related literature. While it is valuable to see that this pattern is reconfirmed in the domain of machine unlearning, the result itself is not very surprising.

---

> ### Author Rebuttal · Authors · 2026-03-31
>
> **Weakness 1, Q1**: Add more data sets and unlearning methods.
>
> **Response**:To test the generalizability of our conclusions, we additionally evaluate two widely used unlearning methods, Negative Preference Optimization (NPO) and Gradient Ascent (GA), using Gemma 2 under the same experimental setup as in the paper. Both methods show patterns consistent with those observed under DPO unlearning, providing evidence that our conclusions hold across diverse and widely-used unlearning methods. Due to the rebuttal character limit, we report partial results for cross-lingual transfer (Table 1) and steering vectors (Figure 2) below, and would be happy to provide the full results during the discussion stage if helpful. This table is interpreted in the same way as Table 1 in the paper.
>
> Cross-lingual Transfer pattern (NPO):
>
> |Family|Unlearn|Query-EN|Query-CH|Query-DE|Query-RU|Query-TU|
> |---|---|---|---|---|---|---|
> |FT-En|Base|97|12|86|47|31|
> ||EN|-85|-6|-73|-34|-19|
> ||CH|-28|-4|-30|-22|-10|
> ||DE|-35|18|-50|-17|0|
> ||RU|-35|10|-36|-28|3|
> ||TU|-46|11|-38|-21|-21|
> |FT-Tu|Base|19|6|18|15|91|
> ||EN|-14|1|-5|-5|-60|
> ||CH|10|0|28|2|-20|
> ||DE|-6|15|-13|15|-26|
> ||RU|1|21|24|-10|-12|
> ||TU|-5|14|7|10|-77|
>
> GA:
> |Family|Unlearn|Query-EN|Query-CH|Query-DE|Query-RU|Query-TU|
> |---|---|---|---|---|---|---|
> |FT-En|Base|97|12|86|47|31|
> ||EN|-92|-6|-68|-32|-22|
> ||CH|-35|-2|-40|-18|-8|
> ||DE|-61|14|-48|-10|0|
> ||RU|-68|15|-55|-22|-13|
> ||TU|-46|6|-46|-15|-15|
> |FT-Tu|Base|19|6|18|15|91|
> ||EN|-3|8|-9|-1|-79|
> ||CH|25|0|16|0|-42|
> ||DE|0|12|-7|3|-74|
> ||RU|8|13|8|-2|-64|
> ||TU|-3|24|8|29|-69|
>
> **Steering Vectors**:
> For NPO, the steering vector recovers 61% of the forgotten knowledge on average (Min=51% Turkish; max=69% English), and is most effective around layer 26 across languages. For GA, average recovery is 59% (min=42% German; max= 81% Chinese), with the strongest effect around layer 20. These results are consistent with the DPO findings reported in Figures 2 and 8 of the paper.
>
> These additional results show that our findings generalize across unlearning algorithms. Similarly, the fact that steering-based recovery remains effective across methods supports our broader conclusion that current unlearning methods suppress, rather than erase, knowledge We appreciate your valuable suggestions and will include complete results in the revision.
>
> **Weakness 2**: Validation of the translation quality and NLI-based answer scores.
>
> **Translation quality**:
>
> During human annotation of the NLI scores, annotators were also asked to inspect whether the translated samples preserved the meaning of the original English examples and whether there were any issues in translation. Across the 250 reviewed TOFU samples (50 per language), we did not observe any translation inaccuracies. We believe this is partly because TOFU examples are short and fact-based, making them easier to translate than more context-dependent tasks. We will discuss this more explicitly in the revision.
>
> **Metrics Verification**:
>
> To address this comment, we added two additional evaluation metrics beyond the NLI score: ROUGE-L recall, a widely used word-overlap metric in unlearning [1,2,3], and GPT-5 as an LLM judge as an approximate upper bound. We then evaluated how well these two metrics align with human judgment on 50 samples each in EN, DE, and CH (omitting RU, TU due to time constraints but can report results in the discussion period if requested). We use the same examples as in paper Table 7 and compare against the reported NLI agreement:
>
> |Metrics|EN|CH|DE|
> |---|---|---|---|
> |LLM%|96|94|94|
> |ROUGE%|66|62|68|
> |NLI%|86|90|84|
>
> Human agreement for LLM and NLI are both very high, while ROUGE is (expectedly) lower as it over-relies on word overlap, missing correct answers that paraphrase the ground truth. LLM was both time-intensive and costly to run. Given their comparable performance, we argue that NLI is an adequate metric to use.
>
> We agree that additional metrics strengthen the credibility of our results. We will report all metrics as above in the paper.
>
> [1]  https://openreview.net/forum?id=B41hNBoWLo.
> [2] https://arxiv.org/pdf/2505.15214
> [3] https://arxiv.org/pdf/2404.05868
>
> **Weakness 3**:The finding that language similarity affects cross-lingual transfer is not particularly new in the related literature.
>
> **Response**:Please refer to our response to Weakness 3 from Reviewer x46o, due to the word limit. Thank you.
>
> **Question2**: Multilinguality of Qwen and Gemma
>
> **Response**:Qwen2.5-7B officially supports over 29 languages (including those in our paper; see model card [1]). While the training data of Gemma2-9B is primarily English, it has been shown to perform competitively on a variety of multilingual benchmarks (e.g., Table 4 in [2]). Our two chosen models therefore represent explicitly supported multilinguality (Qwen) and weaker multilinguality (Gemma). We will clarify this in the paper.
>
> [1] https://huggingface.co/Qwen/Qwen2.5-7B
> [2] https://arxiv.org/pdf/2412.15115

---

> > ### Author Rebuttal · Reviewer_mtH3 · 2026-04-02
> >
> > Thank you for your responses. I appreciate the authors' following actions, but I guess the revision (e.g., introducing more metrics to different experiments) would require the amount of work that may not be covered by minor modifications possible in the camera ready.
> > So, let me maintain my score.

---

> > > ### Author Response · Authors · 2026-04-03
> > >
> > > Thank you for acknowledging that our response has fully addressed your concerns. We would like to clarify that the required revisions to the paper are indeed minimal.
> > >
> > > Regarding metrics, we proposed adding the metric-validation table above, rather than rerunning all experiments with the two additional metrics. As the results show, ROUGE has relatively low alignment with human judgment, so reporting it throughout the paper would introduce misleading conclusions. For the LLM judge, the differences relative to NLI are also small. For example, the averaged scores for the results reported above changes by <<10% (EN: .45->.48; CH: .38->.42; DE: .44->.50) when using LLM judgment instead of NLI. To further illustrate the effect on the experiments, we also reproduced the first shuffling configuration of fine-tuning in English and unlearning in English across five query languages from Table 1, and again observed only small changes (EN: .10 -> .14; CH: .13 -> .17; DE: .03 -> .05; RU: .07 -> .08; TU: .06 -> .05). These differences are negligible and do not affect any of the trends in the main paper.
> > >
> > > Similarly, all other additional results are already precomputed. The additional unlearning results will be added to the appendix and prominently linked from the main paper. All remaining updates will be incorporated into the main paper using the additional page allowed for camera-ready submissions.
> > >
> > > We would appreciate if you could reconsider your scores in the light of this additional information.

---

### Decision · Program_Chairs · 2026-04-30

**Decision:**

Accept (regular)

**Comment:**

The idea of unlearning is remove sensitive knowledge encoded in LLMs. Much of the focus of unlearning research has been on English. The reviewers acknowledge that this is the first work which does a cross-lingual analysis and comes up with interesting findings including that unlearning depends upon the similarity between languages and that in most cases, knowledge is supressed rather than removed and can be recovered using steering vectors.

The paper received 1 R, 1 WR, 1 WA and 1 A with the most confident reviewer recommending acceptance. Two reviewers questioned whether the findings will be applicable to different unlearning methods. In their rebuttal the authors extended their analysis to two unlearning methods but the reviewer concluded that this would constitute a major revision of the paper.

Given that the most confident reviewer (4) recommends acceptance, I am recommending a weak accept